# Time-to-recovery after cesarean section delivery among women who gave birth through cesarean section at Hawassa University Comprehensive Specialized Hospital, South Ethiopia: A prospective cohort study

**Anteneh Fikrie** [1,2]*, **Rahel Zeleke**[2], **Henok Bekele**[2], **Wongelawit Seyoum**[2], **Dejene Hailu**[3], **Zelalem Jabessa Wayessa** [4], **Girma Tufa**[4], **Takala Utura** [1], **Male Matie**[5], **Gebeyehu Dejene Oda** [6]

1 School of Public Health, Institute of Health, Bule Hora University, Bule Hora, Southeastern Ethiopia, 2 Departement of Public Health, Pharma College Hawassa Campus, Hawassa, Ethiopia, 3 School of Public Health, College of Medicine and Health Sciences, Hawassa University, Hawassa, Southern Ethiopia, 4 Midwifery Department, Institute of Health, Bule Hora University, Bule Hora, Southeastern Ethiopia, 5 SNNP Regional Health Bureau Director, Disease Prevention Health Promotion Directorate, Hawassa, South Ethiopia, 6 Policy Planning, Monitoring and Evaluation, JSI Research and Training Institute SNNP Regional Coordinator, Hawassa, South Ethiopia

* antenehfikrie3@gmail.com

**Data Availability Statement:** Data essential for the conclusion are included in this manuscript.

## Abstract

Cesarean deliveries have become a major public health problem worldwide in recent decades. In addition, information on the quality of service, as measured by timely recovery is scarce. This study was assessed predictors of recovery time after cesarean section among women who delivered by cesarean section at Hawassa University Comprehensive Specialized Hospital (HU-CSH) Southern Ethiopia. Institution-based prospective cohort study design was conducted among 381 study participants from July to August 2020. A consecutive sampling technique employed to select study participants. A pre-tested structured questionnaire was used to collect the data. The data were entered and analyzed by Epi info version 7 and SPSS respectively. Bivariable and multivariable Cox regression used to identify the predictors of time-to-recovery after cesearean section. Adjusted Hazard Ratio (AHR) with the respective 95% confidence intervals (CIs)and p-value <0.5 was used to declare statistical significance. A total of 369 mothers who undergone cesearean section were followed for 1,042 person-days of observation. The timely recovery (within 4 days) was found to be 96.2% [95%CI: 94.04–98.4%] and the overall median (IQR) time of recovery was 2.00 (2, 3) days. The study revaled that the Incidence density rate (IDR) of timely recovery was found to be 0.34 per person-days or 2.38 per person-week. Whereas, the cumulative probability of not recovered on the 1st and 4th day was 0.995 and 0.038 respectively. This study found that women who had ANC follow-up (AHR = 1.49, 95%, CI: 1.05–2.10) and discharge from the wound site (AHR = 0.13, 95%, CI: 0.03–0.56) were identified as a significant positive and

**Funding:** Pharma College has funded the research to AF. The funder had no part in study design, information gathering, and analysis, judgment to publish, or development of the manuscript.

**Competing interests:** The authors have declared that no competing interests exist.

**Abbreviations:** ACOG, American College of Obstetrician and Gynecologists; AHR, Adjusted Hazard Ratio; BMI, Body Mass Index; CDs, Cesarean Delivery; CI, Confidence Intervals; CHR, Crude Hazard Ratio; CS, Cesarean Section; HIV, Human Immune Deficiency Virus; HUCSH, Hawassa University Comprehensive Specialized Hospital; IQR, Inter Quartile Range; PROM, Premature Rapture of Membrane; SPSS, Statistical Package for Social Science; SSI, Surgical Site Infection; WHO, World Health Organization.

negative predictors of time-to-recovery after CS delivery respectively. The rate of early recovery obtained by this study was comparable to the global level figures. Still, the cleanness of the surgical site to prevent the incidence of postsurgical site CS delivery is very essential.

## Introduction

Cesarean section(CS) is the most commonly performed surgical procedure worldwide that effectively prevents maternal and newborn mortality when used for medically indicated reasons [1–3]. However, there is a lack of evidence revealing its benefits for women who do not require the procedure [2]. Despite the progressively increased CS rates worldwide over the last decades; still, the trend has not been accompanied by significant maternal or perinatal benefits. Conversely, the existing evidence showed that higher rates of CS have been associated with increased maternal and perinatal morbidity [1].

For more than three decades the ideal rate for cesarean sections was estimated to be between 10% and 15% [2]. However, nearly 20% of births were delivered by CS universally of which the vast majority (more than 40%) was observed in Latin America and the Caribbean followed by Oceania and North America with a prevalence rate of 35%. In England, 27.8% of all pregnant women undergo CS to deliver their babies [4]. Likewise, in Ethiopia most recently a prevalence rate of 24.7% was reported [5]. Many factors had been implicated for the substantial rise of the CS rates [3] such as an increase in the prevalence of obesity, multiple pregnancies, an increase in the proportion of nulliparous women or older women, fear of pain, physician factors, increasing fear of medical litigation, as well as organizational, economic and social factors [1].

An increasing rise in the rate of CS delivery becomes a major public health concern. This is because CS delivery has associated with short- and long-term health consequences in comparison to vaginal delivery for women, babies and their households [3, 6]. According to a large study result in low and middle-income countries; one-fourth of women were died after giving birth through CS [3]. CS is associated with short-term risks such as; blood transfusion, the risks of anesthesia complications, organ injury, infection, and long-term risks that can extend many years beyond the current delivery and affect the health of the woman, the child, future pregnancies [1, 2, 7, 8].

Despite the rise in rates, cesarean delivery is the single most important factor associated with 5–20 folds increased postpartum infection than vaginal delivery [4, 9, 10]. Likewise, studies conducted in Ethiopia found that 11–15% of women who had given birth through CS developed surgical site infection which later leads to delayed recovery [10, 11]. Moreover, information on the quality of the service, as measured by timely recovery is scarce, particularly in the study area. Therefore, this study assessed time-to-recovery after cesarean section delivery and predictors among women who gave birth through cesarean section at Hawassa University Comprehensive Specialized Hospital (HU-CSH), southern Ethiopia.

## Materials and methods

### Ethics statement

The ethical Committee that approved this study was Institutional Review Board of Pharma College. The reference number was P/C/H/C/240/13. An additional official letter of

corroborating was also obtained from the HUCSH Chief executive Office. All the participants were approached immediately to delivery by the data collectors and invited to participate in the study voluntarily and took informed verbal consent from each mother before data collection. The ethics committees had approved the verbal consent procedure. Code numbers was used in place of identifiers to maintain the confidentiality of participants' information. Moreover, information regarding any specific personal identifiers like the name of the participants was not collected, and also the confidentiality of any personal information was also maintained. All methods were performed in accordance with the relevant ethical guidelines and regulations.

## Study design and settings

The institution-based prospective cohort study design was conducted from July to August 2020 at Hawassaa University Comprehensive Specialized Hospital (HU-CSH), Southern Ethiopia. The hospital is located in Hawassa city, the capital of Sidama regional state. It is one of the teachings and specialized and comprehensive hospitals in the Southern part of Ethiopia. It has over 400 beds. The average monthly CS deliveries are estimated to be 200. There are 72 midwives nurses and 7 senior Gynecologist doctors serving in the hospital.

## Study population and sample size

The source population was all women who gave birth through CS in HU-CSH. In this study we used an internal comparisons group. So, we have categorized study participants in terms of their specific exposure status (ANC visit Yes/No, Hypertension Yes/No, Diabetes mellitus Yes/No, Previous history of CS Yes/No, Types skin incision Transverse/Vertical . . .etc. . ..) and then compared the time-to-recovery after cesarean section delivery in exposed individuals and compared with their counter parts. The sample size was calculated based on the double population proportion formula by using Epi info version 7 [12] computer program considering the following assumptions: 95% CI, power 80%, the ratio of unexposed to exposed 1:1 and percent of outcome in exposed group 24.63% and percent of outcome in unexposed group 12.2% [11], a Risk ratio of 2 and 10% non-response rate. Finally, the calculated sample became 381. Finally, mothers who underwent cesarean section delivery at the study hospital were selected using consecutive sampling method.

## Data collection procedure and data quality control

Three trained midwives from Labor and Obstetrics and Gynecology ward, and a Public Health Officer were participated in the data collection and supervision, respectively. Both the data collectors and supervisors were trained for two days on the procedures of data collection. A pretested structured questionnaire which was adapted by reviewing different peer-reviewed articles were used to collect the data [5, 13]. Data like socio-demographic characteristics (age, marital status, residences, BMI, educational status, religion, family income, number of children, source of referral), medical and obstetrics characteristics (ANC visits, number of ANC visits, HTN, DM, HIV/AIDs, anemia, blood transfusion, history of abortion and gestational age during CS), Pre, Intra and post-operative characteristics (Pervious history of CS, number of CS delivery, Types skin incision, breastfeeding, type of anesthesia used, presence of discharge from wound site, mobility after CS delivery), the date of CS procedure done and discharge date were obtained by the face-to-face interview, individual maternal records and referral notes review.

## Data processing and analysis

The data were thoroughly cleaned, coded, and then entered into Epi info version 7 and exported to the Statistical Package for Social Science (SPSS) version 20 for analysis. Descriptive analysis was run to assess missing values and the presence of outliers. The dependent variable was time-to-recovery after CS delivery. The recovery time after CS delivery was dichotomized into early recovered or censored. Those who stayed ≤4 days after the CS delivery were taken as early recovered [14]; whereas the others were regarded as censored (late recovered) based on their length of stay. Length of stay (LOS) is the number of days the women stayed in the hospital from the date CS was done until the women develop an event of interest (early recovery) or censored (late recovery). Length of stay was computed using the difference between the date of discharge and the date of CS procedure done.

A multicollinearity test was carried out to see the correlation between the independent variables and no multicollinearity between independent variables was witnessed (Variance inflation factor <10). Data were described using frequency distribution and measures of central tendency and dispersion. Kaplan- Maier Curve and Long rank test was used to estimate cumulative survival probability and to compare survival status probability across different groups. Cox proportional-hazard regression was used to adjust the potential cofounding variables and identify predictors of time-to-recovery. Variables with a *p*-value less than 0.25 during bivariablee analysis were considered as a candidate for multivariable analysis to check multicollinearity effects. The assumption of proportional hazard was graphically evaluated by the log-minus-log survival curve. Adjusted hazard ratio (AHR) with 95% confidence interval (CI) were used to present the output of the analysis [15].

## Results

### Socio-demographic characteristics of participants

From a total of 381 women delivered by CS, the data of 369 with a response rate of 96.8%. Almost all, 365 (98.9%) of women were married. The majority 307 (83.2%) of the women came from urban areas. The mean (SD) age of the participants was 27(3.5) years and the majority (90.2%), were less than the age of thirty years. The majority of women, 127 (34.4%) and 193 (52.3%) have college and above level of education and housewife in their occupation respectively. Nearly half, 176 (47.7%) of the participant's family have a monthly income of 63.06–126.13$ range. More than three-fourths, 277(76.2%) of the women have the source of referral, and the majority, 205(74%) of them were referred from health centers (**Table 1**).

### Medical and obstetrics characteristics of the women

The vast majority of participants, 328(88.9%) had at least one ANC visit. Accordingly, 31.1% and 57.7% of the women had 1–3 and ≥ Four ANC visits respectively. The mean (±SD) Gestational age (GA) during delivery was 37.9 (±1.72) weeks and 24.7% had GA of fewer than 37 weeks. Regarding chronic diseases; about 3.5%, 2.7%, and 1.9% of the women had hypertension, diabetes mellitus and HIV/AIDs respectively. Likewise, nearly one in five, 19.2% of women have been diagnosed with anemia and 1.6% of them received a blood transfusion (**Table 2**).

### Pre, intra, and post-operative characteristics of the women

One-in-seven, (15.4%) of the women had a previous history of CS delivery; whereas the majority (84.4%) of them had one-time exposure including the current. The vast majority 94.6% of

**Table 1. Socio-demographic characteristics of the women on predictors of time-to-recovery from cesarean section delivery among women who gave birth by CS at HUCSH, 2020 (n = 369).**

| Variable | | Frequency | Percent (%) |
|---|---|---|---|
| Age in years | ≤20 | 22 | 6 |
| | 21–25 | 99 | 26.8 |
| | 26–30 | 212 | 57.5 |
| | ≥31 | 36 | 9.8 |
| Educational level | No formal education | 26 | 7 |
| | Primary education | 94 | 25.5 |
| | Secondary education | 122 | 31.1 |
| | College and above | 127 | 34.4 |
| Religion | Orthodox | 110 | 29.8 |
| | Protestant | 186 | 50.4 |
| | Muslim | 73 | 19.8 |
| Residence | Urban | 307 | 83.2 |
| | Rural | 62 | 16.8 |
| Occupation of mother | Housewife | 193 | 52.3 |
| | Private employee | 58 | 15.7 |
| | Governmental employee | 79 | 21.4 |
| | Others@ | 39 | 10.6 |
| Family monthly income in USD | 63.06$ | 42 | 11.4 |
| | 63.06–126.13$ | 176 | 47.7 |
| | 126.13–252.27$ | 125 | 33.9 |
| | 252.27$ | 26 | 7 |
| Number of children | <2 | 259 | 70.2 |
| | 2–3 | 80 | 21.7 |
| | ≥4 | 30 | 8.1 |
| Do you have source of referral | Yes | 277 | 76.2 |
| | No | 92 | 23.8 |
| The source of referral (n = 277) | Health Centre | 205 | 74 |
| | Hospital | 72 | 26 |
| BMI (kg/m2) | Normal | 264 | 71.5 |
| | Overweight | 105 | 28.5 |

@ Merchant, student, unemployed (n = 369).

women have undergone transverse type skin incision. About 3% of the women experienced discharge from the wound site (**Table 3**).

## Time to recovery from cesarean section delivery

In this study, the proportion of timely recovery (within 4 days) is 96.2% [95%CI: 94.04–98.4%]. The overall median (IQR) time-to-recovery was 2.00 (2, 3) days. The overall incidence density rate (IDR) of timely recovery was calculated using person-days of follow-up as a denominator for the entire cohort. Thus, 369 study participants were followed for 1,042 person-days of observation. Henceforth, the IDR is 0.34 per person-days or 2.38 per person-week. Whereas, **the cumulative probability of not recovered** on the 1st and 4th day was 0.995 and 0.038 respectively. The overall mean survival time was 3.07(95%CI: 2.75–3.40) days (**Table 4**).

**Table 2. Medical and obstetrics characteristics of the women on predictors of time-to-recovery from cesarean section delivery among women who gave birth by CS at HUCSH, 2020 (n = 369).**

| Variable | Category | Frequency | Percent |
|---|---|---|---|
| ANC visit | Yes | 328 | 88.9 |
| | No | 41 | 11.1 |
| Number of visits (n = 328) | No visits | 41 | 11.1 |
| | 1–3 times | 115 | 31.2 |
| | ≥Four times | 213 | 57.7 |
| Hypertension | Yes | 13 | 3.5 |
| | No | 356 | 96.5 |
| Diabetes mellitus | Yes | 10 | 2.7 |
| | No | 359 | 97.3 |
| HIV/AIDs | Yes | 7 | 1.9 |
| | No | 362 | 98.1 |
| Anemia | Yes | 71 | 19.2 |
| | No | 298 | 80.8 |
| Blood transfusion | Yes | 6 | 1.6 |
| | No | 369 | 98.4 |
| History of abortion | Yes | 51 | 13.8 |
| | No | 318 | 86.2 |
| Number of abortions (n = 51) | One time | 45 | 88.2 |
| | ≥ two times | 6 | 11.8 |
| GA at during CS | <37 weeks | 91 | 24.7 |
| | 37–40 weeks | 251 | 68 |
| | >40 weeks | 27 | 7.3 |

**Table 3. Pre, intra and post-operative characteristics of the women on predictors of time-to-recovery from cesarean section delivery among women who gave birth by CS at HUCSH, 2020 (n = 369).**

| Variables | Category | Frequency | Percent (%) |
|---|---|---|---|
| Previous history of CS | Yes | 57 | 15.4 |
| | No | 312 | 84.6 |
| Number of CS done including the current | One time | 312 | 84.4 |
| | Two time | 46 | 12.5 |
| | ≥ Three time | 11 | 3 |
| Types skin incision | Transverse | 349 | 94.6 |
| | Vertical | 20 | 5.4 |
| Breastfeeding | Yes | 352 | 95.4 |
| | No | 17 | 4.6 |
| Type of anesthesia used | General | 23 | 6.2 |
| | Local | 346 | 93.8 |
| Discharge from the wound site | Yes | 11 | 3 |
| | No | 358 | 97 |
| Bad odour discharge | Yes | 9 | 2.4 |
| | No | 360 | 97.8 |
| Mobility after CS | Yes | 367 | 99.5 |
| | No | 2 | 0.5 |
| Indication for CS | Maternal | 185 | 50.1 |
| | Fetal | 184 | 49.9 |

**Table 4. Life table analysis of severely among women who gave birth by CS delivery at HUCSH, 2020.**

| Interval Start Time | Number Entering Interval | Number Withdrawing during Interval | Number Exposed to Risk | Number of Terminal Events | Proportion Terminating | Proportion Surviving | Cumulative Proportion not recovered at End of Interval |
|---|---|---|---|---|---|---|---|
| 0 | 369 | 0 | 369.000 | 0 | .00 | 1.00 | 1.00 |
| 1 | 369 | 0 | 369.000 | 2 | .01 | .99 | .99 |
| 2 | 367 | 0 | 367.000 | 214 | .58 | .42 | .41 |
| 3 | 153 | 0 | 153.000 | 116 | .76 | .24 | .10 |
| 4 | 37 | 0 | 37.000 | 23 | .62 | .38 | .04 |
| 5 | 14 | 1 | 13.500 | 0 | .00 | 1.00 | .04 |
| 6 | 13 | 1 | 12.500 | 0 | .00 | 1.00 | .04 |
| 7 | 12 | 0 | 12.000 | 0 | .00 | 1.00 | .04 |
| 8 | 12 | 1 | 11.500 | 0 | .00 | 1.00 | .04 |
| 9 | 11 | 0 | 11.000 | 0 | .00 | 1.00 | .04 |
| 10 | 11 | 0 | 11.000 | 0 | .00 | 1.00 | .04 |
| 11 | 11 | 2 | 10.000 | 0 | .00 | 1.00 | .04 |
| 12 | 9 | 3 | 7.500 | 0 | .00 | 1.00 | .04 |
| 13 | 6 | 0 | 6.000 | 0 | .00 | 1.00 | .04 |
| 14 | 6 | 2 | 5.000 | 0 | .00 | 1.00 | .04 |
| 15 | 4 | 1 | 3.500 | 0 | .00 | 1.00 | .04 |
| 16 | 3 | 1 | 2.500 | 0 | .00 | 1.00 | .04 |
| 17 | 2 | 1 | 1.500 | 0 | .00 | 1.00 | .04 |
| 18 | 1 | 0 | 1.000 | 0 | .00 | 1.00 | .04 |
| 19 | 1 | 1 | .500 | 0 | .00 | 1.00 | .04 |

### Factors associated with an early recovery time of women delivered by CS

Multivariable Cox regression was carried out for variables verified as significant at p = value, < 0.25 during bivariate Cox regression. Accordingly, after adjusting for different variables ANC follow-up and discharge from the wound site were found to be independent predictors of recovery time in women delivered by CS at HU-CSH. Accordingly, women who had ANC follow-up were 1.5 times more likely to recover early as compared to their counterparts (AHR = 1.49, 95%, CI: 1.05–2.10). On the other hand, women who had discharge from the wound site had an 87% reduced chance of early recovery time than those women who did not have discharge from the wound site (AHR = 0.13, 95%, CI: 0.03–0.56) (Table 5).

On the other hand, time-to-recovery patterns of the women delivered by CS across selected variables were compared using the Log-rank test. Hence, there were significantly different recovery rates among women with and without PROM. The mean recovery time with the presence and absence of PROM in women was 7.66 and 2.86 days respectively and their difference was statistically significant (Log-rank test = 18.659, $P<0.001$). Similarly, women who received general and spinal anesthesia (Log-rank test = 25.663, $P<0.001$) and women who had and had not discharge from the wound site (Log-rank test = 48.623, $P<0.001$) had a statistically significant difference in the recovery times (Table 6).

## Discussion

This study was conducted to assess the time-to-recovery after cesarean section delivery and predictors among women who gave birth through cesarean section at Hawassa University Comprehensive Specialized Hospital (HU-CSH), southern Ethiopia. The finding of this study

**Table 5. Output of bivariable and multivariable Cox regression analyses on factors associated with time-to-recovery after CS delivery among women who gave birth by CS delivery at HUCSH, 2020.**

| Variables | Outcome | | CHR | AHR |
|---|---|---|---|---|
| | Early recovered N (%) | Censored N (%) | (95% CI) | (95% CI) |
| Age in years | | | | |
| ≤30 | 326(91.8) | 7 (50) | 1 | 1 |
| ≥31 | 29 (8.2) | 7 (50) | 0.67 (0.46–0.99) | 0.90 (0.60–1.33) |
| Residence | | | | |
| Rural | 57 (16.1) | 5(35.7) | 0.97(0.73–1.29) | - |
| Urban | 298(83.9) | 9(64.3) | 1 | - |
| Occupation | | | | |
| Employed | 131 (39.9) | 6 (42.9) | 1 | |
| Unemployed | 224 (63.1) | 8(57.1) | 1.00(0.80–1.24) | |
| Source of referral | | | | |
| No | 87 (24.5) | 1(7.1) | 1.13(0.89–1.44) | 1.02(0.79–1.31) |
| Yes | 268 (73.6) | 13 (92.9) | 1 | 1 |
| ANC visit | | | | |
| Yes | 317 (89.3) | 3 (21.4) | 1.78(1.26–2.51) | 1.49(1.05–2.10)* |
| No | 38 (10.7) | 11(78.6) | 1 | 1 |
| Prolonged labour | | | | |
| Yes | 105 (29.6) | 2 (14.3) | 1.08(0.86–1.36) | |
| No | 250 (70.4) | 12 (85.7) | 1 | |
| Previous abortion | | | | |
| Yes | 49 (13.8) | 2 (14.3) | 1.03(0.76–1.39) | |
| No | 306 (86.2) | 12 (85.7) | 1 | |
| Anemia | | | | |
| Yes | 67 (18.9) | 4 (28.6) | 0.85(0.65–1.11) | 0.98(0.74–1.29) |
| No | 288 (81.1) | 10 (71.4) | 1 | 1 |
| Gestational age at CS | | | | |
| Term | 245 (69) | 2 (14.3) | 1.24(0.99–1.56) | 1.13(0.89–1.12) |
| Pre & post term | 110 (31) | 12 (85.7) | 1 | 1 |
| Previous CS | | | | |
| Yes | 47(13.2) | 10 (71.4) | 0.71 (0.52–0.97) | 0.87 (0.64–1.20) |
| No | 308 (86.8) | 4 (28.6) | 1 | 1 |
| Discharge wound site | | | | |
| Yes | 2 (0.5) | 9 (64) | 0.09(0.024–0.38) | 0.13(0.03–0.56)** |
| No | 353 (99.5) | 5 (36) | 1 | 1 |
| Types skin incision | | | | |
| Transvers | 339 (95.5) | 10 (71.4) | 1.59(0.97–2.59) | 1.55(0.58–4.09) |
| Vertical | 16 (4.5) | 4 (28.6) | 1 | 1 |
| Type of anesthesia used | | | | |
| General | 15 (4.2) | 8 (57.1) | 0.49(0.30–0.81) | 1.31(0.46–3.71) |
| Local | 340 (95.8) | 6 (42.9) | 1 | 1 |
| Chronic disease | | | | |
| Yes | 42 (11.8) | 12 (85.7) | 0.70 (0.50–0;97) | 0.92 (0.66–1.29) |
| No | 313 (88.2) | 2 (14.3) | 1 | 1 |
| PROM | | | | |
| Yes | 4 (1.1) | 5 (35.7) | 0.29(0.11–0.79) | 0.65(0.23–1.83) |
| No | 351 (98.9) | 9(64.3) | 1 | 1 |

** & * show statistically significant association at P < 0.01 and 0.05 respectively.

**Table 6. Log rank (Mantel-Cox) test for association of explanatory variables with time-to-recovery after CS delivery among women who gave birth by CS delivery at HUCSH, 2020.**

| Variables | Mean recovery time(95% CI) | Overall comparison Log Rank | |
|---|---|---|---|
| | | $X^2$ | *P*-value |
| PROM | | | |
| Yes | 7.66 (4.49–10.83) | 18.659 | <0.001 |
| No | 2.86 (2.59–3.14) | | |
| Age | | | |
| ≤30 | 2.75 (2.52–2.98) | 11.357 | 0.001 |
| ≥31 | 5.69 (3.55–7.83) | | |
| Discharge from the wound site | | | |
| Yes | 15.90 (12.03–19.78) | 48.623 | <0.001 |
| No | 2.62 (2.46–2.79) | | |
| Previous CS | | | |
| Yes | 4.98 (3.53–6.42) | 12.767 | <0.001 |
| No | 2.67 (2.44–2.88) | | |
| Gestational age at CS | | | |
| Term | 2.55 (2.38–2.72) | 10.15 | 0.001 |
| Pre & post-term | 4.09 (3.21–4.97) | | |
| Chronic disease | | | |
| Yes | 5.98 (4.12–7.84) | 12.445 | <0.001 |
| No | 2.56 (2.42–2.71) | | |
| Type of skin incision | | | |
| Transverse | 2.94 (2.61–3.21) | 11.120 | 0.001 |
| Vertical | 5.55 (3.01–8.06) | | |
| Type of anesthesia | | | |
| General | 8.26 (5.05–11.47) | 25.663 | <0.001 |
| Local | 2.66 (2.48–2.84) | | |
| ANC Visit | | | |
| Yes | 2.56 (2.39–2.73) | 32.139 | <0.001 |
| No | 6.30 (4.38–8.22) | | |

revealed that the overall proportion of timely recovery (within 4 days) after a maximum of 19 days is 96.2% with the median (IQR) time of recovery being 2.00 (2, 3) days.

The finding of this study is similar to the WHO-recommended average stay of 3–4 days in hospital after a CS delivery [14]. The result of this study is in line with other similar studies conducted in Ethiopia: Butajira and Attat hospitals [13]. However, the result is lower than a large population-based study conducted in North-Eastern Italy, where the recovery time was 4.7 days [16] and in India, the recovery time was 8.6 days [17]. The most likely reason for the similarity of the recovery rate might be due to the participant's socio-demographic characteristics, and the health care providers who performed the CS. Moreover, the hospitals are serving as teaching institutions and lower rates of most comorbid conditions. Four years back a study conducted at the same hospital found that amongst the total mothers who underwent CS, 65 (11.0%) developed surgical site infection [1]. However, in our study, only 11 (3%) of women were developed surgical site infections. This shows the progress of quality service delivery of the hospital.

On the other hand, our study found that the median (IQR) time of recovery time was 2.00 (2,3) days. This is corroborated by studies conducted outside Ethiopia, where the average of the women was discharged within 2 days [14, 18]. However, the finding of this study was

inconsistent with other studies conducted in Ethiopia: Butajira and Attat hospitals where the mean recovery time was 3.27 [13] and a study conducted in 30 low and mid-income countries showed that the mean (±SD) hospitalization after the cesarean section was 5.9 (±3.4) days in the studied localities [19]. This implies that the study hospital has an improved, and quality of pre, intra, and post-operative services which help the women to recover early. Moreover, all the women who underwent CS at the study hospital were discharged alive; this indicates that the quality of the procedure was at an optimal level.

In this study, the mean time-to-recovery among women whose ages were ≤30 and ≥31 years was 2.75 and 5.69 days respectively (Log-rank test = 11.357, $P$ = 0.001). The same finding was reported by different studies where younger women were discharged earlier [3, 11, 18, 19]. This might be because increased age has been associated with different comorbidities which affect the length of stay at the hospital. Similarly, women who had had and had not had chronic disease had a statistically significant difference in the recovery times (Log-rank test = 12.445, $P$<0.001). This result is supported by different studies; the hospital stay of women with complications and comorbidities was longer [1, 16, 18, 19]. This is due to the reason that women with complications and/or with co-morbidities need additional services for the management. So, this might prolong her length of stay at the hospital. Moreover, the study hospital is serving as a referral for the surrounding and adjacent woredas and zones of the Oromia region and Gedio Zones catchment populations. This could overestimate the number of complicated mothers.

Our study revealed that the women who had discharge from the wound site had 87% reduced recovery time than those women who did not have discharge from the wound site (AHR = 0.13, 95%, CI: 0.03–0.56). This finding is supported by studies conducted in Ethiopia [1, 11]. Another study conducted in England found that women undergoing CS are at higher risk of developing postnatal infection and this makes the recovery time longer [4]. Evidence suggested that the occurrence of surgical site infection is expected to increase as the incidence of CS increases [9]. The magnitude of wound infection (2.9%) found in this study hospital is comparable to the global guidelines for the prevention of surgical infection (2.9%) following cesarean section delivery [6]. This implies improvements in hygiene conditions, antibiotic prophylaxis, sterile procedures, and other practices in our study hospital. However, still, women undergoing a CS had better be equipped with pertinent information on how to keep the surgical site clean, post-operative recovery, and infection prevention advice.

Use of ANC designed to guide and support women on the mode of birth after a primary cesarean delivery is advantageous [20]. Accordingly, our study found that women who had ANC follow-up were 1.5 times more likely to recover early as compared to their counterparts (AHR = 1.49, 95%, CI: 1.05–2.10). This result is by supported another study that reported, ANC and correctly indicated CS can positively impact on health outcomes of the mother [21]. The possible reason might be because during ANC follow-up women with a previous cesarean birth, chronic diseases, and women with pregnancy-related complications could be identified and the decision for the mode of delivery could be agreed upon based on the woman' preference.

This study has added weight to the existing literature by quantifying the time-to-recovery following cesarean section delivery; as a result it could be used as an input for policymakers and health program developers on maternal health services. Nevertheless, the findings from this study would be difficult to infer to the wider population, because the study was a hospital, HUCSH is a tertiary type of hospital which is serving as a teaching institution for different disciplines including specialties. So, this might positively affect the quality of the services given to the mothers unlike that of the general and primary hospitals where scare obstetrics and gynecologists are found. Moreover, the sample was relatively small; some of the variables, such as

qualification of the person who performed the CS, type of health facility where the CS was performed as we have included the referral cases were not assessed.

## Conclusions

The rate of early recovery and the median (IQR) time of recovery obtained by this study were comparable to the global level figures. The overall incidence density rate (IDR) of recovery in the cohort was 0.34 per Person-days or 2.38 per person-week. Women who had ANC follow-up and discharge from the wound site were identified as a positive and negative predictor of time-to-recovery after CS delivery respectively. The HUCSH Obstetrics and gynaecology department should stress on women undergoing CS want more information on what constitutes a 'normal' post-operative recovery and keeps the cleanness of the surgical site to prevent the incidence of postsurgical site infection which is the major predictor for time-to-recovery after CS delivery.

## Acknowledgments

We would like to acknowledge Hawassa University Comprehensive Specialized Hospital and Pharma College Hawassa Campus for permitting us to undertake this study. Our thanks also go to our study participants, data collectors, supervisors, and those who were actively participated in our study.

## Author Contributions

**Conceptualization:** Anteneh Fikrie, Rahel Zeleke, Dejene Hailu.

**Data curation:** Anteneh Fikrie, Rahel Zeleke, Dejene Hailu.

**Formal analysis:** Anteneh Fikrie, Rahel Zeleke, Dejene Hailu.

**Funding acquisition:** Anteneh Fikrie, Rahel Zeleke, Dejene Hailu.

**Investigation:** Anteneh Fikrie, Rahel Zeleke, Dejene Hailu.

**Methodology:** Anteneh Fikrie, Rahel Zeleke, Dejene Hailu, Takala Utura, Male Matie, Gebeyehu Dejene Oda.

**Project administration:** Anteneh Fikrie, Rahel Zeleke, Wongelawit Seyoum, Dejene Hailu, Takala Utura, Gebeyehu Dejene Oda.

**Resources:** Anteneh Fikrie, Rahel Zeleke, Wongelawit Seyoum, Dejene Hailu, Takala Utura, Male Matie.

**Software:** Anteneh Fikrie, Rahel Zeleke, Henok Bekele, Dejene Hailu, Zelalem Jabessa Wayessa, Girma Tufa, Takala Utura, Male Matie.

**Supervision:** Anteneh Fikrie, Rahel Zeleke, Henok Bekele, Wongelawit Seyoum, Dejene Hailu, Takala Utura, Male Matie.

**Validation:** Anteneh Fikrie, Rahel Zeleke, Henok Bekele, Wongelawit Seyoum, Dejene Hailu, Zelalem Jabessa Wayessa, Girma Tufa, Takala Utura, Male Matie, Gebeyehu Dejene Oda.

**Visualization:** Anteneh Fikrie, Rahel Zeleke, Henok Bekele, Wongelawit Seyoum, Dejene Hailu, Zelalem Jabessa Wayessa, Girma Tufa, Takala Utura, Male Matie, Gebeyehu Dejene Oda.

**Writing – original draft:** Anteneh Fikrie, Rahel Zeleke, Dejene Hailu, Zelalem Jabessa Wayessa, Girma Tufa, Takala Utura, Male Matie, Gebeyehu Dejene Oda.

**Writing – review & editing:** Anteneh Fikrie, Rahel Zeleke, Henok Bekele, Wongelawit Seyoum, Dejene Hailu, Zelalem Jabessa Wayessa, Girma Tufa, Takala Utura, Male Matie, Gebeyehu Dejene Oda.

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
