## [Decision Letter · Decision Letter 0]

9 Mar 2022

PGPH-D-21-00438

Time-to-recovery after cesarean section delivery among women who gave birth through cesarean section at Hawassa University Comprehensive Specialized Hospital, south Ethiopia: A prospective cohort study

Dear Dr. Fikrie,

Thank you for submitting your manuscript to PLOS Global Public Health. After careful consideration, we feel that it has merit but does not fully meet PLOS Global Public Health’s publication criteria as it currently stands. Therefore, we invite you to submit a revised version of the manuscript that addresses the points raised during the review process.

We look forward to receiving your revised manuscript.

Kind regards,

Charles Anawo Ameh, PhD

Academic Editor

Journal Requirements:

1. In the Methods, please clarify that participants provided oral consent. Please also state in the Methods:

- Why written consent could not be obtained

- Whether the Institutional Review Board (IRB) approved use of oral consent

- How oral consent was documented

For more information, please see our guidelines for human subjects research: https://journals.plos.org/plosone/s/submission-guidelines#loc-human-subjects-research

2. In the Funding Information you indicated that no funding was received. Please revise the Funding Information field to reflect funding received.

Please ensure that the funders and grant numbers match between the Financial Disclosure field and the Funding Information tab in your submission form.

3. Please update your Competing Interests statement. If you have no competing interests to declare, please state: “The authors have declared that no competing interests exist.”

Additional Editor Comments (if provided):

Dear Authors, thanks for submitting a very interesting article, however before I can make a final decision on it, there are several minor corrections to be made and I recommend a detailed proof read including grammar check before resubmitting. Thanks

Reviewers' comments:

Reviewer's Responses to Questions

**Comments to the Author**

1. Does this manuscript meet PLOS Global Public Health’s publication criteria? Is the manuscript technically sound, and do the data support the conclusions? The manuscript must describe methodologically and ethically rigorous research with conclusions that are appropriately drawn based on the data presented.

Reviewer #1: Yes

Reviewer #2: Yes

2. Has the statistical analysis been performed appropriately and rigorously?

Reviewer #1: Yes

Reviewer #2: Yes

3. Have the authors made all data underlying the findings in their manuscript fully available (please refer to the Data Availability Statement at the start of the manuscript PDF file)?

Reviewer #1: No

Reviewer #2: Yes

4. Is the manuscript presented in an intelligible fashion and written in standard English?

Reviewer #1: Yes

Reviewer #2: No

5. Review Comments to the Author

Reviewer #1: This is an interesting study of length of stay post-cesarean section (CS) in a relatively large university hospital in Ethiopia. They describe the overall LoS and seek to find factors associated with short LoS.

Isues:

1. Although indeed this is a prospective study with a cohort of women, this is not a cohort study because there is no clear exposure defined. The outcome is the LoS in days from CS to discharge from the health facility. There is no censoring here. Yes, it could be some administrative censoring on day 4 for those who have LoS beyond that. That is OK. But when you are describing the length of stay do not censor. The cumulative proportion of discharge at day 4 is still the same 96.2% (95%CI: 93.7 to 97.7%). And now the mean has sense to be estimated [with censoring the mean requires some assumptions to be meaningful].

It is OK to proceed with administrative censoring on day 4 for the Cox regression.

2. Why the Kolmogorov Smirnov test for normality. We do not need this.

3. On the results and discussion, it would be better to remove the “±” on the SD.

4. Table 5:

- the “abortion” variable sounds to be “previous abortion” correct?

- Why age was dichotomized? And please use as reference the category with more observations

5. Table 6:

- How the means were computed? Does the method used here account for censoring?

- It seems that these are unadjusted mean recovery times. I imagine computing the adjusted ones but state that these are unadjusted.

- Please do not use chi-squared as a measure of association. Please something as time-ratios or mean-differences with its 95% confidence intervals, please.

6. There are English issues.

- For example, the 3rd paragraph of the background somewhere at the “one-forth of women were died… ”

- Another example… In the paragraph of “data collection procedure and data quality control”

- there are abbreviations that never a full spelled such HTN, DM etc

Reviewer #2: Gross grammatical errors noted. This would have been resolved with a grammar checking app before submission. Revisions attached.

Gross grammatical errors noted. This would have been resolved with a grammar checking app before submission. Revisions attached.

6. PLOS authors have the option to publish the peer review history of their article (what does this mean?). If published, this will include your full peer review and any attached files.

**Do you want your identity to be public for this peer review?** For information about this choice, including consent withdrawal, please see our Privacy Policy.

Reviewer #1: **Yes: **Orvalho Augusto

Reviewer #2: No

---

## [Decision Letter · Decision Letter 1]

22 Jun 2022

PGPH-D-21-00438R1

Time-to-recovery after cesarean section delivery among women who gave birth through cesarean section at Hawassa University Comprehensive Specialized Hospital, south Ethiopia: A prospective cohort study

Dear Dr. Fikrie,

Thank you for submitting your manuscript to PLOS Global Public Health. After careful consideration, we feel that it has merit but does not fully meet PLOS Global Public Health’s publication criteria as it currently stands. Therefore, we invite you to submit a revised version of the manuscript that addresses the points raised during the review process.

The reviewers had some minor suggestions to the revised mansucript which requires attention, in particular they have concerns that some of their previous comments were not addressed.

Furthermore the reviewer has highlighted some confusing and potentially unclear labelling of table columns. To avoid confusion we suggest paying close attention to the reviewers comments and revise accordingly.

Could you please revise the manuscript to carefully address the concerns raised?

We look forward to receiving your revised manuscript.

Kind regards,

Lucinda Shen, MSc

Staff Editor

Journal Requirements:

Additional Editor Comments (if provided):

Reviewers' comments:

Reviewer's Responses to Questions

**Comments to the Author**

1. If the authors have adequately addressed your comments raised in a previous round of review and you feel that this manuscript is now acceptable for publication, you may indicate that here to bypass the “Comments to the Author” section, enter your conflict of interest statement in the “Confidential to Editor” section, and submit your "Accept" recommendation.

Reviewer #1: All comments have been addressed

2. Does this manuscript meet PLOS Global Public Health’s publication criteria? Is the manuscript technically sound, and do the data support the conclusions? The manuscript must describe methodologically and ethically rigorous research with conclusions that are appropriately drawn based on the data presented.

Reviewer #1: Yes

3. Has the statistical analysis been performed appropriately and rigorously?

Reviewer #1: Yes

4. Have the authors made all data underlying the findings in their manuscript fully available (please refer to the Data Availability Statement at the start of the manuscript PDF file)?

Reviewer #1: Yes

5. Is the manuscript presented in an intelligible fashion and written in standard English?

Reviewer #1: Yes

6. Review Comments to the Author

Reviewer #1: The authors did address some of my previous comments.

1. In the response letter, the authors state that they removed the Kolmogorov Smirnov test for normality. They did not. It is very clear in the 3rd line of the “data processing and analysis”.

2. Please make sure it is written Epi Info (not EPINFO), and please add a citation. Did you use indeed two different versions (in the population size version 7; and the analysis some another version).

3. In the “time to recovery from cesarean section delivery” subsection, in the last 3 lines. It is written, “Whereas, the cumulative probability of early recovery at the 1st and 4th day was 0.995 and 0.038 respectively.” This is mistaken. I believe the confusion starts from the unclear labelling of the columns in table 4. The last column “cumulative proportion surviving at end of interval”, although in the survival analysis language is indeed survival, to avoid confusion I suggest to call it “cumulative proportion of not recovered”. Remember, in this analysis your outcome is positive (whereas in a typical survival analysis it is a negative outcome, and we would show a cumulative survival). If the authors want to report the “cumulative proportion of recovered” do the complementary:

1 - 1, 1 - 0.99, 1 - 0.41, 1 - 0.10, 1 - 0.04 … ie 0, 0.01, 0.59, 0.90, 0.96. So by first day 1% revered, whereas by 4th day 96% recovered.

4. The first 2 citations are reports. Right? Please make sure they are correctly written.

5. In the background the first line of the 3rd paragraph please clarify why the increasing rate of CS delivery becomes a major public health concern.

7. PLOS authors have the option to publish the peer review history of their article (what does this mean?). If published, this will include your full peer review and any attached files.

**Do you want your identity to be public for this peer review?** For information about this choice, including consent withdrawal, please see our Privacy Policy.

Reviewer #1: **Yes: **Orvalho Augusto

---

## [Editor Report · Decision Letter 2]

15 Sep 2022

Time-to-recovery after cesarean section delivery among women who gave birth through cesarean section at Hawassa University Comprehensive Specialized Hospital, South Ethiopia: A prospective cohort study

PGPH-D-21-00438R2

Dear Mr. Fikrie,

We are pleased to inform you that your manuscript 'Time-to-recovery after cesarean section delivery among women who gave birth through cesarean section at Hawassa University Comprehensive Specialized Hospital, South Ethiopia: A prospective cohort study' has been provisionally accepted for publication in PLOS Global Public Health.

Best regards,

Julia Robinson

Executive Editor